# Integrated Methylome and Transcriptome Analysis between the CMS-D2 Line ZBA and Its Maintainer Line ZB in Upland Cotton

**DOI:** 10.3390/ijms20236070

**Published:** 2019-12-02

**Authors:** Meng Zhang, Liping Guo, Tingxiang Qi, Xuexian Zhang, Huini Tang, Hailin Wang, Xiuqin Qiao, Bingbing Zhang, Juanjuan Feng, Zhidan Zuo, Ting Li, Kashif Shahzad, Jianyong Wu, Chaozhu Xing

**Affiliations:** 1State Key Laboratory of Cotton Biology, Institute of Cotton Research of Chinese Academy of Agricultural Sciences, Key Laboratory for Cotton Genetic Improvement, Ministry of Agriculture, 38 Huanghe Dadao, Anyang 455000, China; zhangmeng910305@163.com (M.Z.); guolp@cricaas.com.cn (L.G.); qitx@cricaas.com.cn (T.Q.); zhangxuexian@caas.cn (X.Z.); tanghn@cricaas.com.cn (H.T.); wanghal@126.com (H.W.); qiaoxiuqin@caas.cn (X.Q.); 18439247312@163.com (B.Z.); fengjuanjuan199206@163.com (J.F.); zuozhidan@163.com (Z.Z.); kashifshahzad85@yahoo.com (K.S.); 2Zhengzhou Research Base, State Key Laboratory of Cotton Biology, Zhengzhou University, Zhengzhou 450000, China; ltwlkq@163.com

**Keywords:** upland cotton, cytoplasmic male sterility, DNA methylation, oxidative phosphorylation, transcriptome

## Abstract

DNA methylation is an important epigenetic modification involved in multiple biological processes. Altered methylation patterns have been reported to be associated with male sterility in some plants, but their role in cotton cytoplasmic male sterility (CMS) remains unclear. Here, integrated methylome and transcriptome analyses were conducted between the CMS-D2 line ZBA and its near-isogenic maintainer line ZB in upland cotton. More methylated cytosine sites (mCs) and higher methylation levels (MLs) were found among the three sequence contexts in ZB compared to ZBA. A total of 4568 differentially methylated regions (DMRs) and 2096 differentially methylated genes (DMGs) were identified. Among the differentially expressed genes (DEGs) associated with DMRs (DMEGs), 396 genes were upregulated and 281 genes were downregulated. A bioinformatics analysis of these DMEGs showed that hyper-DEGs were significantly enriched in the “oxidative phosphorylation” pathway. Further qRT-PCR validation indicated that these hypermethylated genes (encoding the subunits of mitochondrial electron transport chain (ETC) complexes I and V) were all significantly upregulated in ZB. Our biochemical data revealed a higher extent of H_2_O_2_ production but a lower level of adenosine triphosphate (ATP) synthesis in CMS-D2 line ZBA. On the basis of the above results, we propose that disrupted DNA methylation in ZBA may disrupt the homeostasis of reactive oxygen species (ROS) production and ATP synthesis in mitochondria, triggering a burst of ROS that is transferred to the nucleus to initiate programmed cell death (PCD) prematurely, ultimately leading to microspore abortion. This study illustrates the important role of DNA methylation in cotton CMS.

## 1. Introduction

Cotton (*Gossypium hirsutum* L.), as a vital resource for plant fiber and oil, is widely cultivated worldwide [1]. However, its low yield is one of the key factors hindering its further development. As in other crops, the utilization of heterosis can effectively increase yield and improve quality in cotton fiber [2,3,4]. As an economical and ideal pollination system, cytoplasmic male sterility (CMS) plays an important role in the production of hybrid seeds [5], and it has been widely used to facilitate the utilization of heterosis in major crops [6]. Presently, different types of cotton CMS lines, such as DBA/ZBA (CMS-D2) [7,8], Zhong41A (CMS-D8) [9], P30A (interspecific hybridization), [10] and H276A (exogenous gene transfer) [11], have been developed and improved. With the successful construction of three cotton lines, including CMS, maintainer, and restorer, many cotton three-line hybrids have been successfully cultivated in China [12]. Nevertheless, the molecular mechanism of CMS in cotton remains obscure.

Many studies have reported that CMS-associated genes are derived from the mitochondrial genome, and restorer of fertility (*Rf*) genes that control fertility restoration in F_1_ hybrids are located in the nuclear genome [13,14,15]. Several CMS-associated genes have been identified and characterized, such as *WA352* in wild-abortive CMS (CMS-WA) rice [16] and *urf13* in T-cytoplasm maize [17], and growing evidence has supported roles for programmed cell death (PCD) and reactive oxygen species (ROS) in the CMS pathway [9,16,18,19,20,21,22]. Even the biogenesis of jasmonic acid was impaired in Honglian (HL)-type CMS rice [23]. In the cotton CMS-D8 line, altered expression levels of ROS scavenging-related genes were found to be likely to cause male sterility [9]. Similarly, ROS were found to be very likely a key factor leading to male sterility in JA-CMS cotton [21]. Furthermore, numerous differentially expressed genes (DEGs) were identified in the CMS line H276A (*Gossypium barbadense* L.) through a comparative transcriptome analysis, most of which were associated with energy-related metabolic pathways (e.g., the tricarboxylic acid (TCA) cycle, respiratory electron transfer, and oxidative phosphorylation), pentatricopeptide repeat (PPR) proteins, and MYB (v-myb avian myeloblastosis vira I oncogene homolog) transcription factors [11]. Recently, transcriptome sequencing was also used to investigate the cotton CMS line C2P5A, which revealed that genes associated with ROS detoxification enzymes, tapetum proteins, transcription factors, vital metabolic pathways including starch and sucrose metabolism, galactose metabolism, ascorbate and aldarate metabolism, glutathione metabolism, and pyruvate metabolism were differentially expressed in anthers [24]. Although some cotton mitochondrial genomes have been reported so far [25,26], relatively little progress has been achieved in terms of CMS-associated genes in cotton.

DNA methylation, also called “the fifth base”, is a major epigenetic modification that participates in many biological processes, including chromatin conformation, alternative splicing, DNA repair, genomic imprinting, transposon silence, and the regulation of temporal and spatial gene expression [27,28,29,30,31,32,33,34]. Emerging evidence has shown that dynamic changes in DNA methylation may mediate transcriptional variation during male reproductive development [35,36]. Over the past several years, the association of DNA methylation status with CMS or genic male sterility (GMS) has been reported in many plants, such as rice [37], wheat [38], maize [39], cotton [35,40], cabbage [41], and tomato [42]. These observations suggest that disrupted genomic DNA methylation in response to high-temperature stress disturbs sugar and ROS metabolism, ultimately resulting in microspore abortion in GMS cotton [35]. It is not known whether DNA methylation changes observed in GMS may occur during anther development of a CMS-D2 line.

In recent years, next-generation sequencing (NGS) technology has been widely used in biological research, producing a large amount of DNA and RNA data [43]. Whole-genome bisulfite sequencing (WGBS), termed the “gold standard” of DNA methylation research, can allow for the determination of methylation patterns at single-base resolution [27,44,45]. Now, the genome sequences of the diploid cotton species *Gossypium arboreum* (AA, 2n = 2x = 26) [46] and *Gossypium raimondii* (DD, 2n = 2x = 26) [47,48] and the complex allotetraploid cotton species *Gossypium hirsutum* L. (AADD, 2n = 4x =52) [49,50,51,52] have all been completed. Thus, the use of ever-improving high-throughput sequencing technologies and high-quality reference genomes has now ensured the mapping of cytosine methylation at the genomic scale in cotton. However, the genome-wide DNA methylation dynamics and the possible role of DNA methylation in cotton CMS (a widely grown fiber crop worldwide) have not been characterized to date.

In this study, a comparative analysis of the integrated methylome and transcriptome was performed in anthers of the CMS-D2 line ZBA and its isonuclear alloplasmic near-isogenic maintainer line ZB using Illumina sequencing technology. A bioinformatics analysis of differentially methylated DEGs suggested that the “oxidative phosphorylation” pathway may play a crucial role in anther development. Combined with our biochemical data, we propose that DNA hypermethylation is involved in regulating the dynamic balance of ROS production and ATP synthesis to maintain the normal development of cotton anthers. In other words, increased ROS may act as a signaling molecule in the mitochondria or be released into the cytoplasm in some way and then prematurely turn on PCD, ultimately leading to CMS. The epigenetic resources here open up a new pathway for elucidating the molecular mechanisms of CMS in cotton.

## 2. Results

### 2.1. Methylation Landscapes of ZB and ZBA

The representative anther phenotypes of the CMS line ZBA and its near-isogenic maintainer line ZB and 0.5% 2,3,5-triphenyltetrazolium chloride (TTC) staining of pollen grains are shown in Figure 1. Obviously, the stamen filaments and stigma of ZBA were shorter than those of ZB (Figure 1a). The anthers of ZBA were more wrinkled and had no pollen grains (Figure 1a,b), whereas the corresponding anthers in ZB were typically dehiscent to release normal pollen grains (Figure 1a,c).

To explore the potential role of DNA methylation dynamics in anther development, we deciphered cytosine methylation at single-base resolution with high confidence across the whole genome of ZB and ZBA anthers using the WGBS technique (Figure 2a). The results of the WGBS are listed in the Appendix A. The Q30 percentages exceeded 94% in each sample. More than 300 million 125 bp paired-end raw reads comprising 78.38 Gb and 76.60 Gb were generated for ZB and ZBA, respectively, representing >30× of the upland cotton TM-1 reference genome, where the genome size was about 2.5 Gb [49,50] (Appendix A). After trimming adapters and filtering low-quality reads, a total of 309,694,292 and 301,003,201 clean reads were obtained for ZB and ZBA, respectively. More than 74% of these clean reads were successfully mapped to the TM-1 reference genome (Appendix A), which was slightly higher than the rate in a recent study of cotton [53], indicating the high credibility and accuracy of the methods and results in this study. Next, these mapped data were used to retrieve the methylation level (ML) of each cytosine site in the CG, CHG, and CHH contexts.

There were over 700 million cytosines covered in each sample, a number sufficient for further analysis (Figure 2a). Of these, a total of 232,123,389 potentially methylated cytosine sites (mCs) (23.40% at CG sites; 23.22% at CHG sites; 53.37% at CHH sites; and H representing A, T, or C) and 209,955,566 mCs (24.61% at CG sites, 24.39% at CHG sites, and 51.00% at CHH sites) were identified in ZB and ZBA, respectively (Figure 2a). Additionally, the percentages of mC, mCG, mCHG, and mCHH in corresponding cytosine contexts across the whole genome of ZB and ZBA anthers were calculated. We found the overall genomic methylation degree of the mCs was significantly higher in ZB (31.59%) than in ZBA (28.58%), and the MLs in CG, CHG, and CHH contexts also presented similar comparative trends (Figure 2b). We wondered here whether CHH hypermethylation accompanied by CG demethylation in *Arabidopsis* and rice endosperms [54,55] may also occur during anther development. Further calculation of the percentages of DNA MLs in C, CG, CHG, and CHH contexts throughout the 26 cotton chromosomes revealed that hypermethylation in ZB relative to ZBA was mainly due to increased MLs in the CHH context, whereas CG and CHG methylation generally showed a decreasing trend in each chromosome (Appendix A). Moreover, these 26 chromosomes exhibited methylation patterns similar to the whole genome, and the MLs of CG, CHG, and CHH in each chromosome were significantly different between ZB and ZBA samples, indicating that there were some local changes in the DNA MLs (Appendix A).

To further compare the DNA methylation patterns of the two samples in different genomic regions, we also analyzed the methylation profiles within genes, including promoters, exons, introns, and repetitive sequences. Apparently, the CG context was the highest and the CHH context the lowest ML among the three contexts in each gene region (Figure 2c). For CG and CHG contexts, the MLs within the promoter, exon, and intron regions of ZB were significantly lower than in ZBA. However, compared to ZBA, the repeat region of ZB showed slightly higher MLs in all three sequence contexts (Figure 2c).

### 2.2. Differential Methylome Analysis between ZB and ZBA

Differences in DNA methylation between ZB and ZBA samples can be quantitatively characterized by differentially methylated cytosines (DMCs) and differentially methylated regions (DMRs). To investigate the difference between the two samples, we compared the DNA methylomes of ZB and ZBA. We identified more than 2.5 million DMCs in ZB relative to ZBA, 73.23% of which were hypermethylated (hyper-DMCs): 64.05% of the hyper-DMCs occurred in the CHH context, signifying an increase mainly in CHH DNA methylation in ZB relative to ZBA (Figure 3a). A circos plot was used to show the difference in overall MLs between the two samples, presenting especially widespread DNA methylation increases in ZB (Figure 3b).

To further characterize the change in DNA methylation between ZB and ZBA samples, a method based on Fisher’s exact test was used to identify DMRs between two methylomes [56]. A total of 4568 DMRs were identified, and the length of most DMRs was found to be approximately 1000 bp (Appendix A). Remarkably, we found that the overall ML of DMRs in ZB was higher than that of ZBA (Figure 3c). Moreover, distribution statistics of the functional genomic regions associated with DMRs were also performed between the ZB and ZBA samples. Obviously, the most hypermethylated and hypomethylated DMRs occurred in the promoter region. More details are shown in Figure 3d. Furthermore, the genes located in DMRs, which are called differentially methylated genes (DMGs), were also characterized. In total, 2096 DMGs were identified in ZB versus ZBA, of which 2298 hypermethylated DMRs overlapped with 731 hypermethylated genes and 2270 hypomethylated DMRs contained 1365 hypomethylated genes (Figure 3e).

### 2.3. Correlation Between Altered DNA Methylation Patterns and Differential Gene Expression during Anther Development

To investigate whether the DNA methylation changes during anther development were associated with changes in gene expression, transcriptome profiles were generated with three biological replicates on the same materials used for methylome analysis. The FPKM (fragments per kilobase of exons per million fragments mapped) values of ZB and ZBA were evaluated with Pearson’s correlation test, and the average correlation coefficients were 0.889 and 0.941, respectively (Appendix A), which was higher than the correlation coefficient in a previous cotton transcriptome study [57]. In addition, a heatmap analysis of 13 selected cotton reference (housekeeping) genes [58,59,60] with their respective FPKM values was performed using TBtools software [61]. Clearly, the expression values of most housekeeping genes between ZB and ZBA were similar (Appendix A). Overall, these results indicated relatively high correlation between the different replicates in our study, which contributed to the accuracy and reliability of the subsequent quantitative gene expression analysis. Next, we identified differentially expressed genes (DEGs) in ZB versus ZBA. In total, 11,944 upregulated and 8482 downregulated DEGs were identified in ZB relative to ZBA (Figure 4a and Appendix A). We also performed a comparative analysis of MLs and the densities of C-sites in the CG, CHG, and CHH contexts in different gene regions of the up- or downregulated DEGs, including promoter, exon, and intron regions (Figure 4b and Appendix A). Both the promoter and intron regions of the upregulated and downregulated DEGs presented significantly reduced MLs in the CG and CHG contexts, whereas only CHH methylation in the promoter region increased slightly in ZB relative to ZBA (Figure 4b).

To further explore the possible association between DNA methylation changes and changes in gene expression, we first examined overlap between the DMGs and DEGs. As is shown in Figure 4c, a total of 677 nonredundant DEGs associated with DMRs (DMEGs) were identified, including 599 that were widely distributed across the 26 chromosomes of upland cotton and 78 DMEGs on 67 different scaffolds. Of these, 262 genes were hypomethylated with upregulated expression levels, and 106 genes were hypermethylated with downregulated expression levels. On the contrary, 142 upregulated and 179 downregulated genes were hypermethylated and hypomethylated in ZB versus ZBA, respectively (Figure 4d). Subsequently, we comparatively analyzed the differential expression levels of all genes and genes located in hyper- or hypomethylated DMRs, and the results are shown with a boxplot. Surprisingly, both hypermethylated (red box) and hypomethylated (green box) genes exhibited marginally higher expression levels compared with all genes (blue box) (Figure 4e).

### 2.4. GO (Gene Ontology) and KEGG (Kyoto Encyclopedia of Genes and Genomes) Enrichment Analysis of DMEGs

To understand the potential role of DNA methylation changes in cotton anther development, we first performed a gene ontology (GO) analysis of DMEGs. For the 248 hypermethylated DEGs (hyper-DEGs) in ZB versus ZBA, 202 of these genes were annotated to 1350 functional categories, including 729 biological processes (BPs), 195 cellular components (CCs), and 426 molecular function (MFs) (Appendix A). This mainly included BPs such as “ribonucleoside monophosphate metabolic process”, “nucleoside metabolic process”, and “glycosyl compound metabolic process”. Besides, “photosystem I”, “mitochondrial envelope”, and “membrane protein complex” were the three most significant enriched CC terms. However, MFs related to “ribulose–bisphosphate carboxylase activity”, “translation regulator activity”, and “pheromone activity” were important functional GO categories that were involved (Appendix A). Here, 351 of the 441 hypomethylated DEGs (hypo-DEGs) were annotated to 1923 functional categories, including 1113 BPs, 271 CCs, and 539 MFs (Appendix A). Of these, only the BP categories “organic substance metabolic process” and “primary metabolic process”, along with “metabolic process”, had the highest enrichment ratios (Appendix A).

To gain insight into the biological functions of DMEGs in ZB versus ZBA, a KEGG pathway enrichment analysis was also carried out. Among the hyper-DEGs in ZB versus ZBA, “photosynthesis” was the most enriched pathway, followed by “oxidative phosphorylation”, “metabolic pathways”, and “glyoxylate and dicarboxylate metabolism”. The first two pathways were most significantly enriched in ZB versus ZBA, with a corrected *P*-value < 0.05 (Figure 5a and Appendix A). Correspondingly, the hypo-DEGs in ZB versus ZBA were mainly assigned in pathways related to “circadian rhythm plant”, “carotenoid biosynthesis”, “biosynthesis of secondary metabolites”, and “galactose metabolism” (Figure 5b and Appendix A).

### 2.5. DNA Hypermethylation Maintains Oxidative Phosphorylation Homeostasis to Ensure the Normal Development of Cotton Anthers

Subsequently, hyper-DEGs involved in the “mitochondrial envelope” term in ZB versus ZBA were chosen for further analysis. A total of 15 hyper-DEGs were enriched in this GO term, of which nine were downregulated and six were upregulated in ZBA relative to ZB (Appendix A). It is worth noting that the four genes with the largest fold changes (more than two times) were all downregulated in the CMS-D2 line ZBA, including *Gh_A06G0349*, *Gh_A05G3831*, *Gh_A01G0843*, and *Gh_A11G3033*, indicating that DNA hypermethylation may play a crucial role in maintaining the integrity of the mitochondrial envelope (Appendix A).

Previous studies have demonstrated that ROS-dependent cellular metabolism and physiological processes play a crucial role in anther development [16,62,63], and unbalanced ROS metabolism causes male sterility [9]. Hence, we focused on DMEGs involved in the “oxidative phosphorylation” signaling pathway in ZB versus ZBA (Figure 4c and Appendix A). A total of 13 DMEGs were enriched in this pathway, and of these, 10 were found to be hypermethylated and three to be hypomethylated in ZB (Figure 6a and Appendix A).

According to their annotations in the SwissProt protein database and the known homologous genes in *Arabidopsis*, seven of these genes, including *Gh_A01G0418* (NADH dehydrogenase (ubiquinone) 1 alpha subcomplex subunit 9, *GhNDUA9*), *Gh_A03G0734* (NAD(P)H–quinone oxidoreductase subunit 5, *GhNU5C*), *Gh_A10G0789* (NAD(P)H–quinone oxidoreductase subunit K, *GhNDHK*), *Gh_D04G0898* and *Gh_D11G3194* (NAD(P)H–quinone oxidoreductase subunit 1, *GhNU1C*), *Gh_Sca005646G04* (NAD(P)H–quinone oxidoreductase subunit J, *GhNDHJ*), and *Gh_Sca006566G05* (NADH dehydrogenase (ubiquinone) iron–sulfur protein 2, *GhNDUS2*), encode the subunits of mitochondrial electron transport chain (ETC) complex I (Appendix A). The other six genes, including *Gh_A03G0725* (ATP synthase subunit a, *GhATPI*), *Gh_A05G0978* (V-type proton ATPase catalytic subunit A, *GhVATA*), *Gh_D04G1994* (ATP synthase subunit alpha, *GhATPAM*), *Gh_D13G2512* (V-type proton ATPase 16 kDa proteolipid subunit, *GhVATL*), *Gh_Sca010487G02*, and *Gh_Sca010487G03* (ATP synthase subunit beta, *GhATPB*), encode the subunits of mitochondrial ETC complex V (Appendix A).

To investigate the effects of DNA methylation changes on gene expression, we examined the transcript levels of these 13 genes in ZB and ZBA using quantitative real-time polymerase chain reaction (qRT-PCR). Interestingly, all 10 genes associated with DNA hypermethylation in different genomic regions were upregulated in ZB. For the three hypomethylated genes, one promoter hypomethylated gene (*Gh_A01G0418*/*GhNDUA9*) was downregulated in ZB, whereas the other two gene body hypomethylated genes were upregulated (*Gh_A10G0789/GhNDHK*) or indistinguishable (*Gh_D13G2512/GhVATL*) (Figure 6b and Appendix A). In addition, these 13 DMEGs were also used for alignment with the protein-coding genes of the mitochondrial genome [25]. Interestingly, two possible mitochondrial targeted protein-coding genes (*Gh_Sca006566G05/GhNDUS2* and *Gh_D04G1994/GhATPAM*) were downregulated in ZBA relative to ZB (Figure 6b). These hyper-DEGs may have been affected by the CMS-D2 cytoplasm. Our results here suggest that DNA methylation may have a positive regulatory role in the expression of many genes involved in the “oxidative phosphorylation” pathway in anthers. To further confirm this finding, both the H_2_O_2_ and ATP contents in the anthers of ZB and ZBA plants were determined. As expected, the H_2_O_2_ content in ZBA anthers was significantly higher than in ZB (Figure 6c), but the relative levels of ATP in the anthers of ZB and ZBA plants presented an opposite trend (Figure 6d). Therefore, we conclude that DNA hypermethylation may play an important role in maintaining the dynamic balance of ROS production and ATP synthesis during anther development.

## 3. Discussion

### 3.1. Overview of DNA Methylome in Cotton CMS Line Decoded by WGBS

DNA methylation has been recognized as a new regulator of plant growth, development, and stress response [34,64,65,66,67,68]. Some researchers have reported that dynamic changes in DNA methylation are involved in the regulation of photoperiod- and/or thermosensitive GMS in rice [69,70,71] and cotton [35,40]; GMS in wheat [38], cabbage [41], and tomato [42]; and CMS in rice [37] and maize [39]. However, the genome-wide cytosine methylation profile and the possible role of DNA methylation in cotton CMS have not been reported so far.

As a predominantly powerful technology for DNA methylation research, whole-genome bisulfite sequencing (WGBS) can enable the determination of methylation patterns at single-base resolution [45]. Recently, WGBS has been applied to decrypt an increasing number of plant methylomes, ranging from model plants such as *Arabidopsis* [27,44] and rice [68,72,73] to economically important crops such as maize [74,75,76], cotton [35,53,77], soybean [78,79], tomato [65], orange [66], and caster bean [80]. In this study, the global DNA methylation pattern was profiled in anthers of the CMS-D2 line ZBA and its near-isogenic maintainer line ZB using WGBS. This is the first single-base-resolution DNA methylome in the cotton CMS line that has been deciphered and used to study the epigenetic regulation mechanism of male sterility. The genome-wide DNA methylation pattern in anthers was found to be similar but slightly different from a recent methylome analysis in cotton leaves under different treatments [53]. Specifically, we observed higher levels of both CG (65.3–68.7%) and CHG (58.7–61.8%) methylation in cotton anthers (Figure 2b) relative to leaves (58.4–61.7% for mCG and 53.8–56.8% for mCHG), whereas the MLs in the CHH context were similar in anthers (18.8–21.8%) and leaves (19.5–24.2%) [53]. These differences may have been due to the different cotton germplasms and tissues used in these two studies.

### 3.2. Widespread Dynamic DNA Methylation and Its Association with Differential Gene Expression during Anther Development

More mCs and higher MLs were observed among the three sequence contexts in ZB compared to ZBA (Figure 2a,b). The DNA methylation increase in ZB relative to ZBA could have been due to an increased number of total mCs or increased MLs of existing mCs. Further analyses of DMCs showed that the increase in DNA methylation was mainly in the CHH context (Figure 3a). All of the above analyses indicated that CHH was significantly hypermethylated in ZB relative to ZBA. In the endosperm of the model plants of *Arabidopsis* and rice and also during tomato development, CG hypomethylation is often accompanied by local CHH hypermethylation [54,55,65]. However, CHH hypermethylation during sweet orange fruit development and ripening is not accompanied by CG and CHG demethylation [66]. Our results found that CHH hypermethylation at the chromosome level may be accompanied by CG and CHG demethylation during anther development (Appendix A). These results indicate that the dynamic regulation of DNA methylation is critical for normal anther development, even if DNA methylation changes in opposite directions in different plants.

There were no significant differences in the number of hypermethylated (2298) and hypomethylated (2270) regions identified between ZB and ZBA (Figure 3e and Appendix A), which was similar to the DNA methylation pattern reported in cabbage GMS [41]. Consistently with previous studies in photoperiod- and thermosensitive male sterile rice [71] and cabbage GMS [41], more DMR-related genes in ZBA were found to be hypermethylated in most gene regions (Figure 3d,e). A comparative analysis of integrated methylomes and transcriptomes can reveal the relationship between DNA methylation dynamics and gene expression [81]. DNA methylation is generally considered to be a marker of transcriptional repression. However, fruit ripening-induced hypermethylation in citrus is highly correlated with gene activation [66]. In our study, some genes affected by DNA methylation were differentially expressed between ZB and ZBA, of which 396 genes were upregulated and 281 genes were downregulated (Figure 4c,d). This finding suggests an important role of DNA methylation not only in suppressing gene expression but also in the activation of some genes during anther development.

### 3.3. Possible Regulatory Role of DNA Methylation in Cotton CMS

The plant mitochondrial ETC, which contains a series of redox-active electron carriers, is considered to be a major site for intracellular ROS production and ATP synthesis [82]. Superoxide (O_2_^•−^), as a proximal ROS, can be generated in significant quantities by reverse-electron transport at the mitochondrial matrix side of complex I (NADH dehydrogenase) [83] when the mitochondria are not making ATP and consequently have a high ∆*p* (protonmotive force) and a reduced coenzyme Q pool or a high NADH/NAD^+^ ratio in the mitochondrial matrix [83,84]. Conversely, for mitochondria that are actively making ATP and consequently have a lower ∆*p* and NADH/NAD^+^ ratio, the extent of O_2_^•−^ production is far lower [84]. Recent research has shown that the expression of genes involved in redox homeostasis and energy metabolism is significantly modulated by DNA methylation in cotton anthers under high-temperature (HT) stress [35,40]. Our KEGG analysis results also found that hyper-DEGs in ZB versus ZBA were significantly enriched in the “oxidative phosphorylation” pathway (Figure 5a and Appendix A). Interestingly, these hypermethylated genes (encoding the subunits of mitochondrial ETC complexes I and V) were all significantly upregulated in ZB (Figure 6a,b). These results indicate that DNA hypermethylation that mediates oxidative phosphorylation homeostasis may play a crucial role in anther development.

In the rice CMS-WA line, WA352 encoded by the mitochondrial CMS gene accumulates preferentially in the anther tapetum, thereby inhibiting nuclear-encoded mitochondrial protein COX11 (CYTOCHROME C OXIDASE 11) functions in peroxide metabolism, leading to ROS bursts and cytochrome c release, which causes premature tapetal PCD and consequent pollen abortion [16]. In our study, among the 13 DMEGs involved in the “oxidative phosphorylation” pathway, two possible mitochondrial targeted protein-coding genes (*Gh_Sca006566G05/GhNDUS2* and *Gh_D04G1994/GhATPAM*) were downregulated in ZBA relative to ZB (Figure 6b). This result suggests that these two hyper-DEGs may be involved in the regulation of the CMS pathway in cotton.

ROS homeostasis is essential for normal anther and pollen development [40,62,85], whereas excessive accumulation of ROS in anthers leads to cell apoptosis and male sterility [9,35,86,87,88]. In this study, a lower extent of H_2_O_2_ production but a higher level of ATP synthesis was found in maintainer line ZB compared to ZBA (Figure 6c,d). This demonstrates that a ROS burst may occur during anther development in the CMS-D2 line ZBA. This finding is consistent with other cotton sterile lines, such as Zhong41A (CMS-D8) [9] and C2P5A [24]. A decrease in ATP but excessive ROS accumulation was found in a transformant expressing the rice CMS-associated gene *orfH79* [89]. On the basis of previously published results and our findings, we speculate that disrupted DNA methylation in ZBA may disturb the homeostasis of ROS production and ATP synthesis in mitochondria by inhibiting the expression of genes in the “oxidative phosphorylation” pathway, resulting in a burst of ROS that prematurely switches on PCD, eventually leading to microspore abortion. However, the exact molecular mechanisms by which DNA methylation regulates the expression of specific genes involved in the CMS signaling pathway during anther development still require further investigation.

## 4. Materials and Methods

### 4.1. Plant Materials

The cotton CMS line ZBA (with a *Gossypium harknessii* cytoplasm (denoted S)) was developed through consecutive backcross procedures with the maintainer ZB containing a normal fertile upland cotton (AD1) cytoplasm (denoted N) as the recurrent male parent [8,90]. The genotypes of ZBA and ZB were designated as S (*rf1rf1*) and N (*rf1rf1*), respectively. Both ZBA and ZB had a similar nucleus genetic background, but with different cytoplasms, so they were a pair of near-isogenic lines of isonuclear alloplasmic type. All materials were developed and seeds were preserved in the Cotton Heterosis Utilization Laboratory (our research group), the Institute of Cotton Research of the Chinese Academy of Agricultural Sciences (ICR-CAAS). ZBA and ZB were planted in the summer of 2015 at Baibi East Experimental Farm, ICR-CAAS, Anyang, Henan Province, China (36°10′N, 114°35‘E). Crop management practices followed local recommendations.

Our previous cytological observations showed that the male abortion of ZBA occurred roughly at the meiosis stage [91], corresponding with the growth of flower buds to a length of about 3 mm [92]. During the squaring stage, 3-mm-long buds were collected centrally and pooled from 50 representative plants of ZBA and ZB. The anthers were excised and then snap-frozen in liquid nitrogen and stored at −80 °C in a freezer for further use.

### 4.2. DNA and RNA Extraction, Quantification, and Qualification

Total genomic DNA was extracted from anthers according to a modified CTAB (cetyltriethylammnonium bromide) DNA extraction protocol [93]. Total RNA was extracted from the anthers of each sample using a Spectrum™ Plant Total RNA Kit (STRN50, Sigma-Aldrich, Saint Louis, MO, USA) following the manufacturer’s instructions. DNA and RNA degradation and contamination were monitored on 1% (*w*/*v*) agarose gels. DNA and RNA purity was checked using a NanoPhotometer^®^ spectrophotometer (IMPLEN, Westlake Village, CA, USA). The DNA and RNA concentration was measured using a Qubit^®^ DNA and RNA Assay Kit in a Qubit^®^ 2.0 Fluorometer (Life Technologies, Carlsbad, CA, USA). RNA integrity was assessed using an RNA Nano 6000 Assay Kit with a Bioanalyzer 2100 system (Agilent Technologies, Palo Alto, CA, USA).

### 4.3. Bisulfite Treatment and Library Construction

About 5.2 μg of genomic DNA per sample spiked with 26 ng lambda DNA was fragmented by sonication to 200–300 bp with a Covaris S220 sonicator and then end-repaired and adenylated. Unmethylated lambda DNA was used as a control for evaluating the bisulfite conversion rate. Subsequently, cytosine-methylated barcodes were ligated to the sonicated DNA in accordance with the manufacturer’s recommendations. These DNA fragments were treated twice with bisulfite using an EZ DNA Methylation-Gold™ Kit (Zymo Research, Irvine, CA, USA), and the resulting single-strand DNA fragments were PCR-amplified using KAPA HiFi HotStart Uracil + ReadyMix (2×). Next, the library concentration was quantified with a Qubit^®^ 2.0 Fluorometer (Life Technologies) and quantitative PCR, and the insert size was assayed on an Agilent Bioanalyzer 2100 system. Clustering of the index-coded samples was performed on a cBot Cluster Generation System using a TruSeq PE Cluster Kit v3-cBot-HS (Illumina, San Diego, CA, USA) according to the manufacturer’s instructions. Finally, WGBS was conducted with approximately 30× sequencing depth per sample on the Illumina Hiseq 2500 platform (Novogene, Beijing, China), and 125 bp paired-end reads were generated. Image analysis and base calling were performed with an Illumina CASAVA pipeline to generate the final 125 bp paired-end reads. The WGBS raw reads were deposited with the National Center for Biotechnology Information (NCBI) under Sequence Read Archive (SRA) accession numbers SRR10314682 (ZB) and SRR10314680 (ZBA).

### 4.4. DNA Methylation Data Analysis

We first used FastQC to perform basic statistical analyses on the quality of the raw reads. The read sequences produced by the Illumina pipeline in FASTQ format were filtered using Trimmomatic software (version 0.35) [94] to generate high-quality clean reads. Then, the bisulfite-treated clean reads of ZB and ZBA were mapped to the TM-1 genome [50] using Bismark software (version 0.16.1) [95] with the default parameters. The paired-end reads that were uniquely aligned with the genome were reserved for further methylation analysis. The potentially methylated cytosine sites (mCs) were extracted using a “methylation extractor” and transformed to bigWig format for visualization using an IGV browser. To identify true mCs, we modeled the sum ***s***^+^*_i,j_* of the methylated counts as a classic binomial (Bin) random variable with the methylation rate **r***_i,j_*: ***s***^+^*_i,j_*~***Bin*** (***s***^+^*_i,j_* + ***s***^−^*_i,j_*, **r***_i,j_*).

To calculate the methylation level (ML), the whole genome was divided into 10 kb bins with no overlap, and the sum of the methylated and unmethylated read counts in each window was calculated. The ML for each C site shows the fraction of mCs and is defined as ML (C) = reads (mC)/[reads (mC) + reads (C)]. Differentially methylated cytosines (DMCs) were identified using Fisher’s exact test with a false discovery rate (FDR) multiple test correction, and DMCs with a *P*-value < 0.05 and a difference of ML >0.2 between two samples were considered candidate DMCs. Differentially methylated regions (DMRs) were identified using swDMR software (version 1.0.6, http://122.228.158.106/swDMR/), which uses a sliding-window approach with a window size of 1000 bp and a step length of 100 bp. Fisher’s exact test with FDR multiple test correction was used to detect DMRs, in which regions with a corrected *P*-value < 0.05 and a difference of ML >0.1 were considered candidate DMRs. Subsequently, genes located in DMRs, called differentially methylated genes (DMGs), were also characterized.

### 4.5. RNA Sequencing and Data Analysis

Approximately 3 µg of total RNA was used to construct sequencing libraries with an NEBNext^®^ Ultra™ Directional RNA Library Prep Kit for Illumina^®^ (NEB, Ipswich, MA, USA), per the manufacturer’s instructions. The libraries, with three biological replicates per sample, were then sequenced on an Illumina Hiseq 4000 platform (Novogene, Beijing, China), and 150 bp paired-end reads were generated. Transcriptomic raw reads were submitted to the SRA database of NCBI under the following accession numbers: SRR10336139 (ZB1); SRR10336138 (ZB2); SRR10336137 (ZB3); SRR10336146 (ZBA1); SRR10336145 (ZBA2); and SRR10336144 (ZBA3).

Adapters and low-quality reads from the raw data were trimmed using Trimmomatic software (version 0.35) [94]. All remaining clean reads were mapped to the cotton reference genome using TopHat (version 2.0.9) [96]. Cuffdiff (version 2.1.1) was used to calculate the FPKM value of each gene and determine the differential expression using a model based on the negative binomial distribution [97]. Transcripts with an adjusted *P*-value < 0.05 were considered to be differentially expressed genes (DEGs) [98]. The GOseq R package [99] was used for GO enrichment analysis, and KOBAS software [100] was used to test the statistical enrichment of the DEGs or DMEGs in KEGG pathways. GO terms or KEGG pathways with a corrected *P*-value < 0.05 were considered significantly enriched.

### 4.6. Quantitative Real-Time Polymerase Chain Reaction (qRT-PCR) Analysis

For qRT-PCR, a total of 1 μg total RNA was first used for first-strand complementary DNA (cDNA) synthesis using a PrimeScript™ RT reagent Kit for Perfect Real Time (RR037A, Takara, Japan) according to the manufacturer’s instructions. Then, the qRT-PCR analysis was performed using TransStart^®^ Top Green qPCR SuperMix (AQ131, TransGen Biotech, Beijing, China) on a Mastercycler ep realplex instrument (Eppendorf, Hamburg, Germany). Each reaction contained 2 μL cDNA template, 0.4 μL of each primer (10 μM), 2× TransStart^®^ Top Green qPCR SuperMix, and 0.4 μL Passive Reference Dye (50×) with ddH_2_O to make the final volume 20 μL. The reaction conditions were pre-denaturation at 94 °C for 30 s, followed by 42 cycles of denaturation at 94 °C for 5 s, annealing at 58 °C for 15 s, and extension at 72 °C for 20 s. A melting curve was also generated for each sample at the end of each run to determine the specificity of the amplified PCR product. Each gene in each sample was analyzed with three replicates and two technical replicates. The upland cotton *Histone3* (*GhHIS3*) gene was used as an internal control with the primers *GhHIS3*-qRT-F (10 μM) and *GhHIS3*-qRT-R (10 μM), and the relative expression level of genes was calculated using the 2^−ΔΔCt^ method, as described in our previous study [90]. The results were obtained from three biologically independent experiments to ensure the reliability of the assay data. Gene-specific primers for qRT-PCR were designed using Oligo 7 Primer Analysis Software [101], were synthesized commercially (BioSune Biotechnology, Shanghai, China), and are listed in the Appendix A.

### 4.7. ROS and ATP Quantification

Anthers with a bud length of approximately 3 mm from both CMS line ZBA and maintainer line ZB plants were collected in 2 mL tubes for a determination of the ROS and ATP contents, with at least five biological replicates and two technical replicates for each sample. The amounts of ROS were measured using an H_2_O_2_ Detection Kit (Beyotime, Shanghai, China) and are expressed as µM/g fresh anther. ATP content was measured using the luciferin–luciferase method [102] following the protocol of the ATP Detection Kit (Beyotime, Shanghai, China).

### 4.8. Statistical Analyses

Each graphical plot in this study represents the results of multiple independent experiments (*n* ≥ 3), and the values are expressed as the mean ± standard deviation (SD). Statistical significance analyses of gene expression levels were performed using two-tailed unpaired Student’s *t*-tests, and a *P*-value < 0.05 was considered to be statistically significant.

## 5. Conclusions

Through integrated methylome and transcriptome analysis, a total of 4568 DMRs and 2096 DMGs were identified in anthers between the CMS-D2 line ZBA and its near-isogenic maintainer line ZB. Among the DMEGs, 396 genes were upregulated and 281 genes were downregulated. The hyper-DEGs were significantly enriched in the “oxidative phosphorylation” pathway, and these hypermethylated genes (encoding the subunits of mitochondrial ETC complexes I and V) were all significantly upregulated in ZB. Combined with our biochemical data, we propose that DNA hypermethylation is involved in regulating the homeostasis of ROS production and ATP synthesis to maintain the normal development of cotton anthers. Our results help to better understand the possible role of DNA methylation in cotton CMS and will accelerate the study of the molecular mechanisms of CMS in cotton.

## Figures and Tables

**Figure 1 ijms-20-06070-f001:**
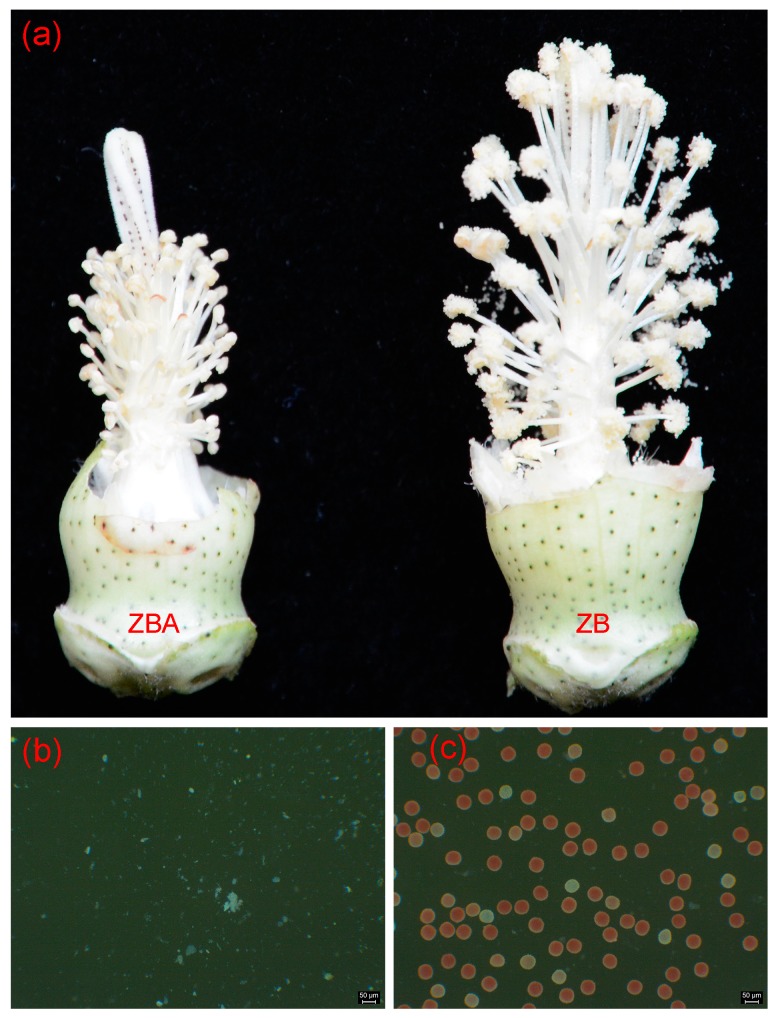
Phenotypic comparison of anthers and pollen grains between the CMS-D2 line ZBA and its near-isogenic maintainer ZB. (**a**) Representative anther phenotypes of ZBA and ZB on the day of anthesis, presenting normal pollen grain release only in ZB. (**b**,**c**) Pollen grains from ZBA (**b**) and ZB (**c**) plants, respectively, stained with 0.5% 2,3,5-triphenyltetrazolium chloride (TTC) on the day of anthesis.

**Figure 2 ijms-20-06070-f002:**
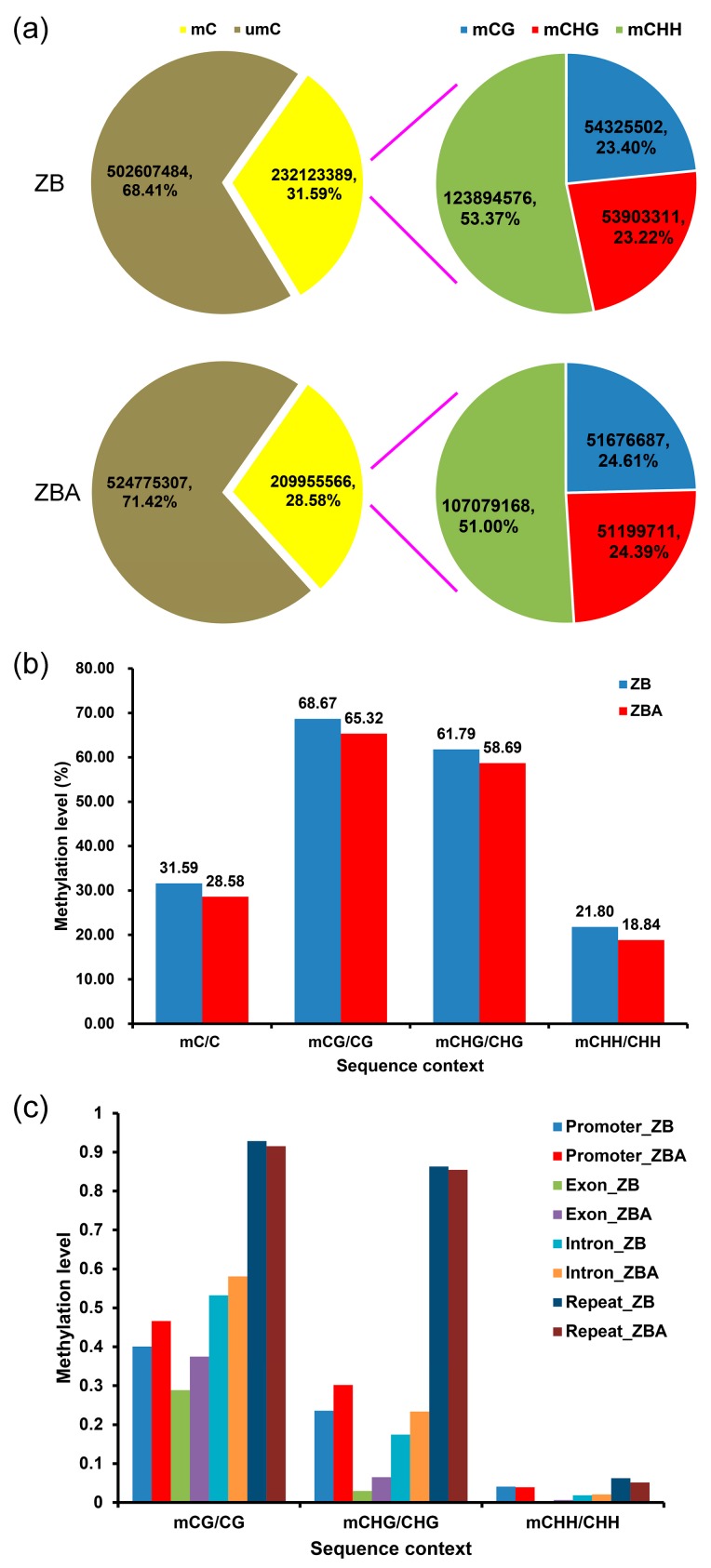
Comparative analysis of DNA methylation patterns between ZB and ZBA. (**a**) Relative proportions of methylated cytosines (mCs) in each sequence context. (**b**) Statistics of methylation levels (MLs) in each sequence context across the whole cotton genome. (**c**) Average MLs in different genomic regions.

**Figure 3 ijms-20-06070-f003:**
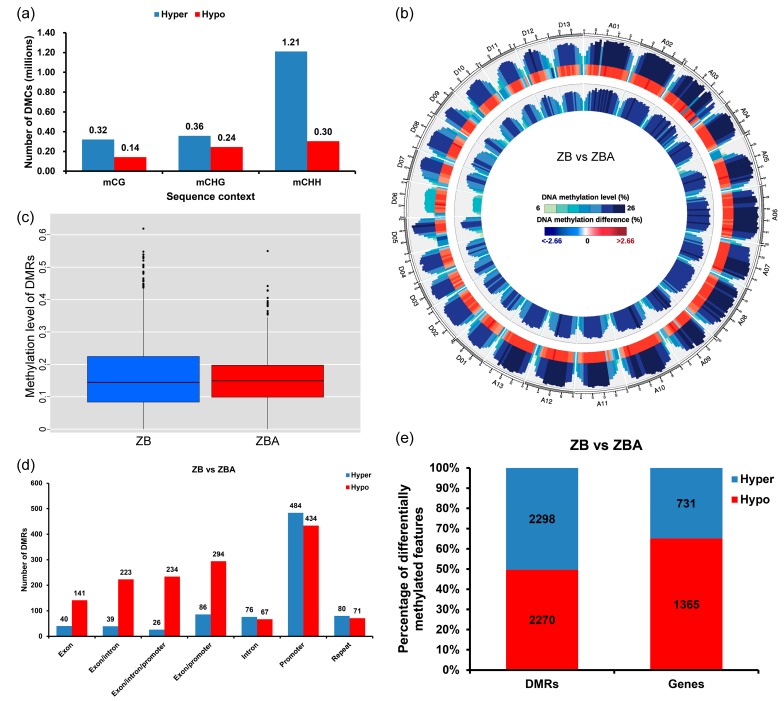
Differential methylome analysis between ZB and ZBA. (**a**) Numbers of differentially methylated cytosines (DMCs) in ZB relative to ZBA are shown for the mCG, mCHG, and mCHH sequence contexts. (**b**) Circos plot showing the difference in overall MLs between the two samples. The outermost rim indicates the chromosome name and scale. The other tracks from outside to inside represent the following: MLs in ZB or ZBA and the difference in overall MLs in ZB versus ZBA. (**c**) Boxplot of the MLs of differentially methylated regions (DMRs) between ZB and ZBA. (**d**) The distribution statistics of the functional genomic regions associated with DMRs. (**e**) Numbers of DMRs and differentially methylated genes (DMGs) in ZB relative to ZBA.

**Figure 4 ijms-20-06070-f004:**
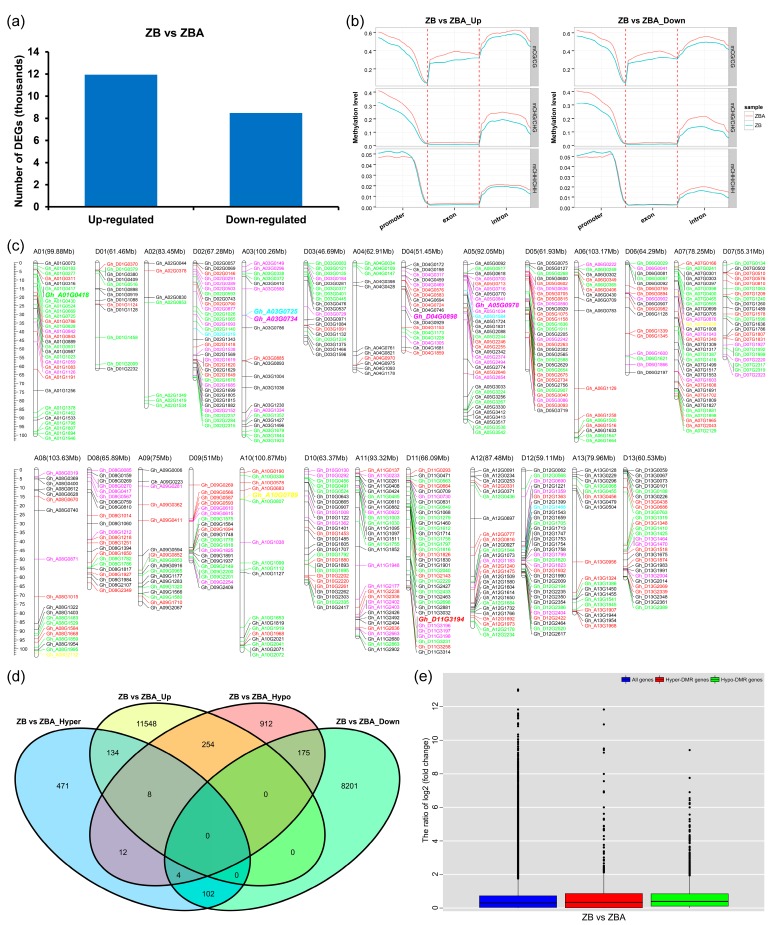
Correlation between altered DNA methylation patterns and differential gene expression. (**a**) Numbers of differentially expressed genes (DEGs) in ZB relative to ZBA. (**b**) Comparative analysis of MLs of C-sites in different gene regions of the up- or downregulated DEGs, including promoter, exon, and intron regions. (**c**) Location distribution of DEGs associated with DMRs (DMEGs) on different chromosomes of upland cotton. The red and black colors represent upregulated DEGs associated with only hyper- and hypomethylated DMRs, respectively. The purple and green colors represent downregulated DEGs associated with only hyper- and hypomethylated DMRs, respectively. The cyan and yellow colors represent up- and downregulated DEGs associated with both hyper- and hypomethylated DMRs, respectively. DMEGs with italicized and enlarged fonts were involved in the “oxidative phosphorylation” pathway. (**d**) Venn diagram showing overlaps between the DMGs and DEGs. (**e**) Boxplot showing the differential expression levels of all genes, hyper-, and hypomethylated genes.

**Figure 5 ijms-20-06070-f005:**
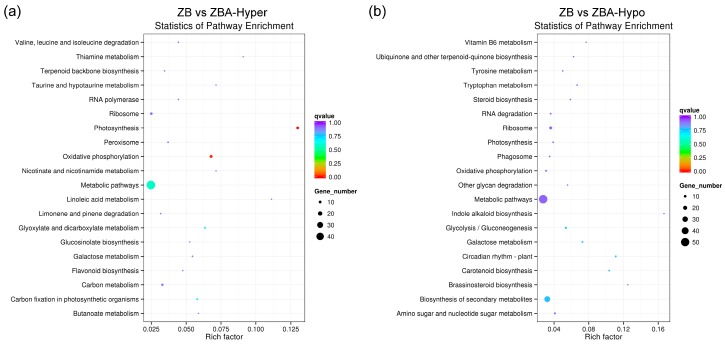
Kyoto Encyclopedia of Genes and Genomes (KEGG) pathway enrichment analysis of hyper- (**a**) or hypomethylated (**b**) DEGs in ZB versus ZBA. The size of the circle represents the number of genes, and the color of the circle represents the *q*-value (corrected *P*-value).

**Figure 6 ijms-20-06070-f006:**
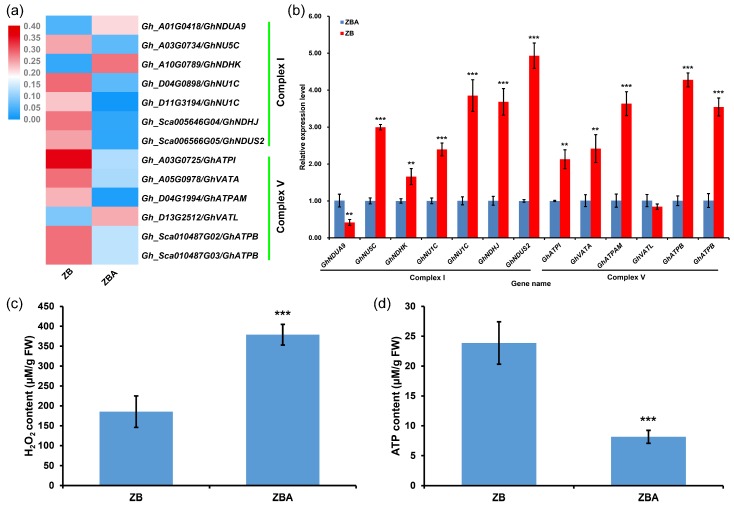
DNA hypermethylation is involved in maintaining oxidative phosphorylation homeostasis during anther development. (**a**) Heatmap of MLs of DMEGs involved in the “oxidative phosphorylation” pathway. (**b**) qRT-PCR validation of relative expression levels of the genes described above (in ZB and ZBA). (**c**,**d**) Detection of H_2_O_2_ (**c**) and ATP (**d**) content in ZB and ZBA anthers. Data (**b**–**d**) are presented as the means ± standard deviation (SD). Vertical bars represent SD of the mean of at least three biological replicates. Asterisks indicate statistically significant differences between ZB and ZBA (** *P* < 0.01; *** *P* < 0.001, Student’s *t*-test).

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
