# Peer review of "Integrated Methylome and Transcriptome Analysis between the CMS-D2 Line ZBA and Its Maintainer Line ZB in Upland Cotton"

_ijms, 2019, doi:10.3390/ijms20236070_

Round 1

Reviewer 1 Report

The manuscript by Zhang et al presents transcriptome and methylome datasets of cotton CMS lines and investigates the molecular mechanism of CMS in cotton. Overall the manuscript is quite descriptive, but clearly written. I have a few suggestions regarding the reliability and validation of data

Is the data submitted to a sequencing repository? The manuscript does not provide any relevant accession numbers. The data should be made publicly available. Also it would allow me to run a few in silico analyses to check the validation of data. The manuscript does not provide information on how many replicates were used for methylone analysis? Looking at the table S1, it seems that the data was not replicated? If the data was replicated, clearly describe in manuscript and provide respective accession numbers. Also include this information in Table S1. It is mentioned that three replicated each were used for transcriptome experiment. Again, respective accession numbers should accompany this information. Also, a phyloenetic analysis should be done on raw reads to illustrate the respective clustering of replicates for each treatment. Authors should provide, in supplement, a table or heatmap of cotton reference (housekeeping) genes with their respective FPKM values. Something similar to what have been provided by other cotton transcriptome papers e.g. Naqvi et al 2017 (Scientific Reports 7, 15880). This would serve an independent verification. To avoid the ambiguity around qPCR analysis, authors should provide a  graph of CT values (generated originally by quantitative thermocycler) for the data presented in Figure 6b

Reviewer 2 Report

The manuscript by Zhang et al. described methylome and transcriptome differences between the CMS-D2 line ZBA and its maintainer line ZB of cotton. Overall, the manuscript reads well, it describes a logical series of experiments and a good job was done on analyzing the obtained data. Several concerns regarding the quality of the manuscript are described below:

The Introduction is too brief. It does not reference previous work related to the genetic, molecular, and biochemical basis of CMS; instead, the authors emphasize methylation as the main cause of CMS. Although the association of methylation with CMS in other plants is cited (lines 57-58), nothing else is mentioned. Please include a paragraph summarizing the current knowledge on CMS mechanisms, to place your work within the context of current knowledge. Please spell out abbreviations (CMS, ETC, DEG) in the abstract. Overall, the experimental details are adequate. It is OK to do a whole genome methylation analysis in the two lines, since nuclear genes may suppress CMS [for example the nuclear restorer of fertility (Rf) genes]. However, it is not clear why the authors did not focus their methylation analysis on mitochondrial DNA, using isolated mitochondria as starting material, since CMS is controlled by mitochondrial genes and is transmitted maternally. At the very least, the authors could have ensured that the sample preparation preserved the integrity of all mitochondrial DNA. The cotton mitochondrial DNA contains a total of 68 genes, including 35 protein-encoding genes (surprisingly, the Liu et al. paper reporting the mitochondrial genome in cotton is not cited in this manuscript). How do the authors know that the DNA of these genes is intact in the final sample prep? In the results section, please include side-by-side comparison of mitochondrial DNA methylation between the two lines. Also, which mitochondrial genes are differentially expressed between ZB and ZBA? Did you find evidence of differential expression of Rf genes (and other relevant genes) between the two lines? Please include this data. While GO and KEGG enrichment analyses are valuable, they are merely a resource for uncovering critical genes for the biological process under investigation. To improve the value of this manuscript, the authors should focus on specific genes, and integrate their findings with existing literature. For example, some hypermethylated DEGs were annotated to the “mitochondrial envelope” CC term. This could be relevant. A mere enumeration of annotations of pathways or processes does not provide sufficient insight. Please discuss the genes involved in the “mitochondrial envelope” and other CC terms, and how they may contribute to better understanding CMS.

Round 2

Reviewer 2 Report

I would like to thank the authors for thoroughly addressing my concerns. This version of the manuscript is considerably improved.